

# Parameter optimization of histogram-based local descriptors for facial expression recognition

Antoine Badi Mame and Jules-Raymond Tapamo

Discipline of Electrical, Electronic, and Computer Engineering, University of KwaZulu-Natal, Durban, KwaZulu-Natal, South Africa

## ABSTRACT

An important task in automatic facial expression recognition (FER) is to describe facial image features effectively and efficiently. Facial expression descriptors must be robust to variable scales, illumination changes, face view, and noise. This article studies the application of spatially modified local descriptors to extract robust features for facial expressions recognition. The experiments are carried out in two phases: firstly, we motivate the need for face registration by comparing the extraction of features from registered and non-registered faces, and secondly, four local descriptors (Histogram of Oriented Gradients (HOG), Local Binary Patterns (LBP), Compound Local Binary Patterns (CLBP), and Weber's Local Descriptor (WLD)) are optimized by finding the best parameter values for their extraction. Our study reveals that face registration is an important step that can improve the recognition rate of FER systems. We also highlight that a suitable parameter selection can increase the performance of existing local descriptors as compared with state-of-the-art approaches.

## INTRODUCTION

Facial expressions are essential for humans to communicate effectively. The most common way humans convey their emotions is by performing facial expressions. This insight into the hidden emotions of humans has led to much attention from the research community to develop facial expression recognition systems. Facial expression recognition (FER) has several applications such as pain assessment (*Rathee & Ganotra, 2015*), health monitoring (*Altameem & Altameem, 2020*), market research (*Garbas et al., 2013*), online learning (*Bhadur, 2021*), to name but a few. A facial recognition system typically operates in three stages: facial detection, feature extraction, and classification.

Feature extraction is one of the key stages in FER because it provides discriminative features related to a particular expression. The features extracted from a facial image must be robust to changes in illumination, pose variation, noise, occlusions, scale variation, and low resolution. The recognition of expressions is also difficult because of the complexity of facial expressions. Local descriptors can solve this problem by capturing the local appearance of image features (such as edges, lines, and spots) related to various facial

Corresponding author
Jules-Raymond Tapamo,
tapamoj@ukzn.ac.za

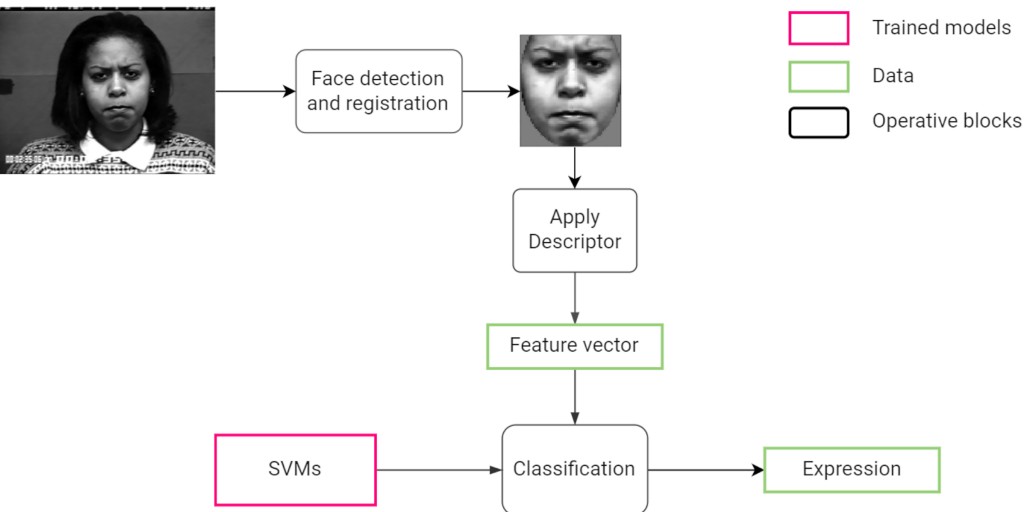

**Figure 1 Proposed facial expression recognition approach.**

expressions. *Jia et al. (2015)* provided a solution for image noise, low resolution, and multi-intensity expressions. *Hesse et al. (2012)* handled different face views, while *Makhmudkhujaev et al. (2019)* studied the recognition of noisy and position-varied facial expressions.

This article investigates four local descriptors which demonstrate powerful characteristics for facial expression recognition. It highlights how a proper parameter selection of these descriptors can increase the efficiency of FER systems in terms of recognition rates and speed. This article further shows the importance of correct face registration to preserve only facial features and discard unnecessary regions.

Figure 1 summarizes the steps followed by the system from processing an input image to the classifying a facial expression.

This work contributes to the field of facial expression recognition in the following ways:

- It proposes a novel face registration algorithm that works for color and gray-scale images.
- It measures the impact of face registration on the efficiency of facial expression recognition through a quantitative analysis.
- It presents a simple method for optimizing the recognition performance of local descriptors. The method is illustrated using four existing local descriptors and evaluated on two publicly available datasets.
- It shows how different feature extraction parameters affect a local descriptor's ability to efficiently represent facial features.

The following is a summary of the remainder of this article: "Background and Related Works" presents the background of this research and some related works. "Materials and Methods" discusses the methodology adopted for facial expression recognition, alongside the datasets used. "Experimental Results and Discussion" assesses the impact of face

registration, and the optimization of local descriptors. The findings and directions for future work are summarized in "Conclusion".

## BACKGROUND AND RELATED WORKS

Extracting robust features from a facial image is one of the most pressing issues for FER systems today. Even the best classifier could fail if the features used do not appropriately represent facial expressions. It has been identified that face registration/pre-processing and feature extraction stages significantly impact the recognition performance of FER systems. The following paragraphs present some developments in the research in these two areas.

### Face registration

Face registration is a procedure that aims to remove image regions that contain no information related to facial expression (such as background and hair) while giving the face a predefined size and pose. Many existing approaches include a normalization or registration step before feature extraction. For instance, *Carcagnì et al. (2015)* used Viola-Jones-based algorithms and ellipse fitting to detect and align the face image while removing background objects. However, the method depends on facial feature color models, limiting the applicability to gray-scale images. *Zhang et al. (2018)* used the Viola-Jones algorithm to detect faces and crop the images to $100 \times 100$ pixels. A similar approach was followed by *Mandal et al. (2019)*. *Hassan & Suandi (2019)* used Cascade Linear Regression to detect facial landmarks. The distance between the eyes was used as a reference to align the face and normalize it $110 \times 150$. *Liu et al. (2021)* used a facial landmark detector that marks 68 points on the face, 27 of which permit the face region to be extracted from the background. Note that this approach does not align the face image.

Based on the above works, face pre-processing plays an essential role in FER systems. However, the impact of face registration on the recognition rate has not sufficiently been quantified. Our work investigates the impact of face registration in more detail. We also propose a novel automatic face registration algorithm inspired by the work in *Carcagnì et al. (2015)* that can also be used with gray-scale images.

### Feature extraction

Robustness of feature extraction is dependent on their discriminative power, the ease with which images are captured, their dimension and tolerance to noise, such as illumination changes, and have low intra-class variations (*Turan & Lam, 2018*). Hence, extensive work has been done to achieve robust feature extraction. Geometric features have been investigated to represent facial expressions (*Ghimire & Lee, 2013*; *Loconsole et al., 2014*; *Lin, Cheng & Li, 2014*). Though geometric features can offer high discriminability, they depend heavily on the accurate detection and alignment of fiducial points in a face. Appearance features can solve this problem because they extract features related to the appearance of the entire face or local patches. *Lu et al. (2015)* proposed combining Gabor features and sparse representation-based classification to recognize facial expressions. The method is robust to illumination changes, different resolutions, and orientation. *Sajjad et al. (2018)* fused two appearance and texture features: Histogram of Oriented Gradients

and Uniform Local Ternary Patterns, and then classified them using support vector machines. Experiments showed that the method performed well in the presence of occlusions. Deep learning methods have also been applied for facial expression recognition. Some notable works include *Zhang et al. (2019)*, *Bougourzi et al. (2020)*, *Dong, Wang & Hang (2021)*. Though deep learning methods can achieve high recognition rates, they often require complex networks and expensive pre-processing computations to produce these results.

Local descriptors have proven to be a powerful way of characterizing expression-related information while being easy to compute. In particular, the histogram-based local descriptors can represent the occurrences of micro-patterns (such as edges, lines, and bumps) that form facial expressions. However, the aggregation of these micro-patterns into histograms leads to a loss of information about the location of the micro-patterns (*Shan, Gong & McOwan, 2009*; *Moore & Bowden, 2011*). To address this issue, weighted histogram representations have been proposed to give higher weights to regions with higher discriminability (*e.g.*, eyes and mouth) and lower weights to regions with lower discriminability (*e.g.*, image borders). *Kabir, Jabid & Chae (2012)* proposed the Local Directional Patterns with variance (LDPv) descriptor to characterize the texture and contrast information of facial components. The novel descriptor adds weights to an LDP (Local Directional Patterns) code based on its variance across the image. Recently, a similar approach called the Weighted Statistical Binary Pattern descriptor was developed by *Truong, Nguyen & Kim (2021)*. The new descriptor combines the histograms of sign and magnitude components of the mean moments of an image. The histograms are weighted based on a new variance moment that contains distinctive facial features. Though the weighted methods achieve high recognition rates and robustness, they can suffer from high dimensionality and implementation complexity.

A popular technique for enhancing a histogram-based local descriptors is dividing the face into overlapping or non-overlapping sub-regions. *Shan, Gong & McOwan (2009)* studied facial expression recognition with the Local Binary Patterns (LBP) descriptor by extracting features from sub-regions of a face image. In this study, an LBP histogram was created with 59 bins and extracted from $18 \times 21$ pixels sub-regions in a $110 \times 150$ image. However, the extraction parameters were not optimized for facial expression recognition. To the best of our knowledge, *Carcagnì et al. (2015)* were the first to optimize the cell size and histogram bins of the Histogram of Oriented Gradients (HOG) descriptor for facial expression recognition. Results show that the best HOG parameters are independent of the datasets used. The study goes further to optimize three other local descriptors and compare them with the HOG descriptor. However, the optimization of the other descriptors is not given in detail, leaving some questions unanswered, such as whether those local descriptors can also be optimized in a way that is independent of the datasets used.

*Benitez-Garcia, Nakamura & Kaneko (2017)* proposed a FER system that extracts appearance and geometric features around specific facial regions. This approach segments three facial regions: eye-eyebrows, nose, and mouth, dividing them into sub-regions. The Local Fourier Coefficients and Facial Fourier Descriptors are extracted from each sub-region, and the feature vectors of all the sub-regions are concatenated. It should be noted

that the resulting feature vector is a linear difference between the expressive and neutral features. Hence, both the expressive and the neutral face are needed to recognize a given expression, which is not always practical in real-life. Also, the approach could be negatively affected by pose variations. *Turan & Lam (2018)* studied the performance of several local descriptors when the resolution and number of sub-regions of the face are varied. Five image resolutions were used, and the face was divided into $l \times l$ pixels sub-images where $l$ was varied from 3 to 11 in steps of 1. The study reports that higher resolutions and more sub-regions lead to better classification performance. However, the results do not show a trend in recognition performance as the number of sub-regions varies. Also, there is not enough discussion on how the resolution and sub-region size affects the descriptor's ability to represent facial features.

Most of the above approaches extract local descriptors by dividing the face images into sub-regions. However, the approaches do not optimize the extraction parameters for FER, except for a few works which attempt to optimize either the sub-block size, image resolution, histogram bins, or a combination of these (*Carcagnì et al., 2015*; *Turan & Lam, 2018*). The current work aims to study the importance of optimizing the sub-block size and histogram bins of local descriptors. We believe that these are the most critical parameters for striking a balance between recognition rate and the feature vector length.

## MATERIALS AND METHODS

### Datasets

Two publicly available face datasets are used in this study and organised as described by *Badi Mame & Tapamo (2022)*. The first one is the CK+ dataset (*Lucey et al., 2010*), which is one of the most used datasets to evaluate FER solutions because it contains subjects from a variety of ethnicities, ages, and genders. This dataset is formed from 123 subjects and is composed of 593 image sequences. These images were captured under controlled and uncontrolled conditions, posed and non-posed statuses. The experiments carried out in this study only used the posed images where the subjects performed six expressions. To obtain a balanced dataset from the available sequences, the following images are selected: the last images in the sequences for anger, disgust, and happiness; the last and fourth images for the first 68 sequences related to surprise; the last and fourth from last images for the images of fear and sadness. The second dataset is the Radboud Faces Database (RFD) compiled by *Langner et al. (2010)*. RFD is a popular dataset among researchers because it contains a variety of expressions, gaze directions, and head orientations. The dataset contains pictures of 67 subjects (both adults and children) displaying eight expressions. A subset was created, selecting 402 images labeled with six expressions, including anger, disgust, fear, happiness, sadness, and surprise. Figures 2 and 3 give some image samples from the CK+ and RFD datasets, respectively. Table 1 summarizes the datasets.

### Face detection and registration

An important step in recognizing a facial expression is to locate a face within the input image. Many robust face detectors have been developed (*Yang, Kriegman & Ahuja, 2002*; *Viola & Jones, 2004*). In the present study, a general frontal face detector is sufficient since

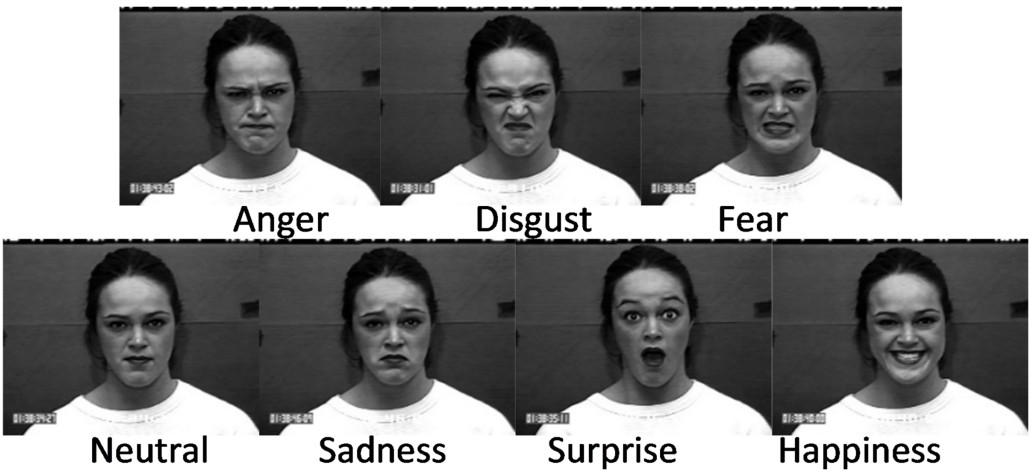

**Figure 2** **Sample of images from the CK+ dataset.**

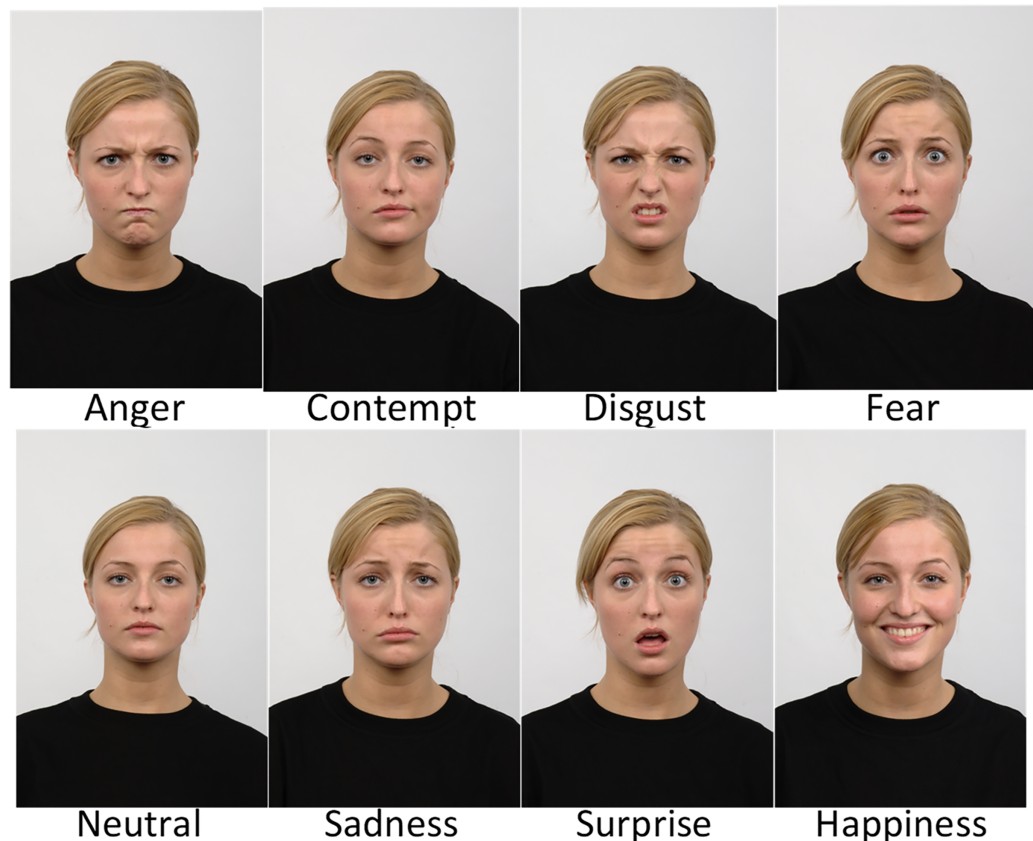

**Figure 3** **Sample of images from the Radboud Faces database.**

recognition is done on frontal faces only. The face detector is based on the Viola-Jones detector implemented by the OpenCV library. A face region contains all the information necessary to observe a facial expression. However, much of this information can be discarded to allow the model only to use the parts of the image containing facial

**Table 1 Summary of datasets.**

| Dataset | No. of images | Number of expressions |
|---------|---------------|----------------------|
| CK+ | 347 | 6 |
| RFD | 402 | 6 |

components. Face registration aims at giving detected faces a predefined pose, shape, and size. According to *Carcagnì et al. (2015)* the best features are usually extracted from images where the eyes are in a consistent position relative to the image. To address this issue, we proposed a novel face registration algorithm that can be used on gray-scale images. First, we enhanced the image by applying contrast limited automatic histogram equalization a gray-scale image. Next, we located the face blob by performing binary segmentation followed by two filtration techniques (see details further below). Given that binary segmentation is a process that separates an image into two regions based on an interval, the challenge here was to find the correct interval within which the face pixels lie. We propose an algorithm to automatically select the best thresholds quickly and efficiently. Consider an image with pixel intensity $I(x, y)$, the segmented image $B$, is defined as

$$B(x, y) = \begin{cases} 1 & \text{if } T_1 < I(x, y) < T_2 \\ 0 & \text{otherwise} \end{cases} \tag{1}$$

where $T_1$ and $T_2$ are the lower and upper thresholds respectively. These thresholds are defined as

$$T_1 = \mu - \sigma \tag{2}$$
$$T_2 = \mu + 2\sigma \tag{3}$$

where $\mu$ and $\sigma$ are the mean and standard deviation of the intensity values respectively. Specifically, the mean and standard deviation are obtained by creating a histogram from the face center pattern. The mean and standard deviation are, respectively, defined by Eqs. (4) and (5).

$$\mu = \frac{1}{N} \sum_{x,y} W(x, y) \times I(x, y) \tag{4}$$

$$\sigma = \sqrt{\frac{1}{N} \sum_{x,y} (I(x, y) - \mu)^2} \tag{5}$$

where $N$ is the number of pixels in an image, and $W(x, y)$ is the frequency of the bin (or interval) where the intensity $I(x, y)$ lies.

The binary segmentation operation is not perfect hence unwanted regions must be filtered out. The first filter is an ellipse which is computed from the face's dimensions and location. Elements inside the ellipse are retained while elements outside the ellipse are discarded. The second filter consists of searching for the largest connected components (*Bolelli et al., 2019*) of the graph-encoded binary image. The blob's shape is then refined by estimating the polygonal shape around it. Specifically, a polygon is estimated by finding a

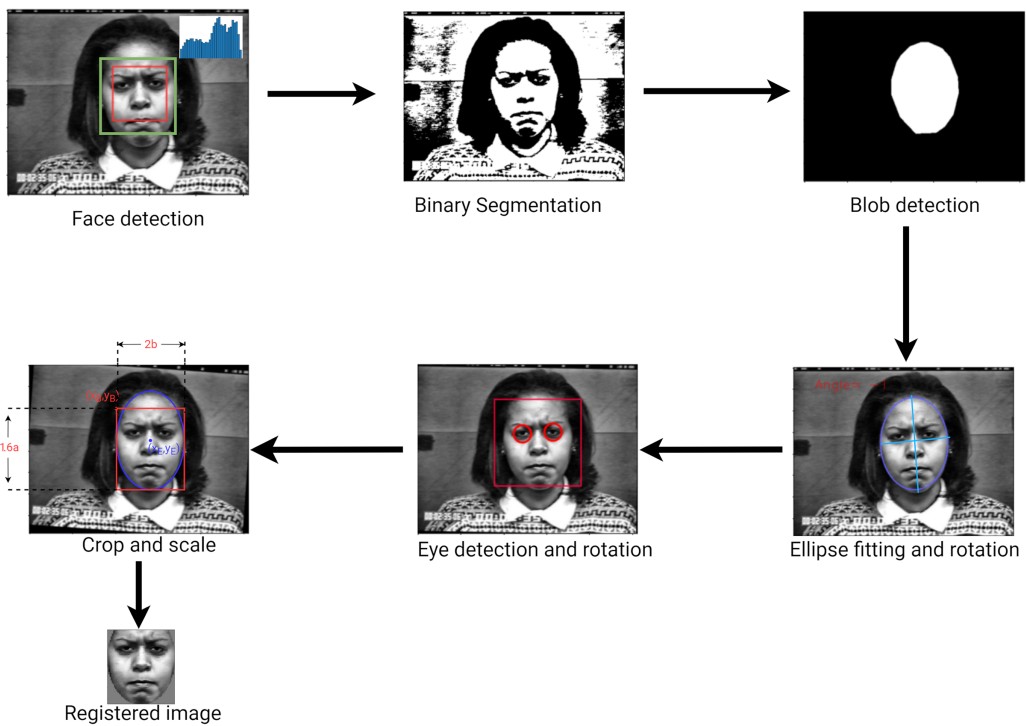

**Figure 4  Face detection and registration approach.**

set of contour points (*Lim, Zitnick & Dollár, 2013*) enclosing the blob, computing its convex hull (*Sklansky, 1982*) and approximate a polygonal shape. Polygon estimation is important because it fills any holes in the blob and makes the ellipse-fitting step more reliable.

A vertical rotation of the face is carried out, by fitting an ellipse around the face and using its angle of inclination (*i.e.*, the angle between the minor axis and the x-axis). To refine the process, the image is rotated a second time using the eye locations. Elliptical cropping followed by rectangular cropping are then performed to generate an image in which eyes are in predefined position relative to the sides of the image. Finally, the output image is resized. The approach used to perform the face detection and registration is shown in Fig. 4.

## Feature extraction

The goal of feature extraction is to generate a feature vector from a facial image for classification. This study is interested in local descriptors used for facial expression recognition. Four local descriptors were examined: Histogram of Oriented Gradients (HOG), Local Binary Patterns (LBP), Compound Local Binary Patterns (CLBP), and Weber's Local Descriptor (WLD). As previously introduced in *Badi Mame & Tapamo (2022)*, this study uses a method that divides the face into a number of non-overlapping and equally-sized sub-blocks, and a local descriptor is extracted from each sub-block, producing several histograms using Eq. (6). The resulting feature vector is formed by normalizing the histograms and concatenating them into a 1D feature vector using Eq. (7).

Histograms of sub-blocks are calculated separately and normalized such that the frequencies sum to one. This rule is applied to all the types of descriptors except the HOG descriptor, where the blocks are normalized in groups using L2-Hys (*Dalal & Triggs, 2005*). Suppose $I_j$ is a matrix, of dimension $W_S \times W_S$, of intensities that represents the $j$-th sub-block in the image, and the function $D_j(x, y)$ calculates the descriptor at a position $(x, y)$ in $I_j$, then the histogram associated with this sub-block is defined as

$$H^j = \{h_i^j\}_{i=0,1,\ldots,N_b-1} \tag{6}$$

$$h_i^j = \sum_{x=0}^{W_S-1} \sum_{y=0}^{W_S-1} \delta(D_j(x, y), i)$$

where $\delta(d, i)$ is defined as

$$\delta(d, i) = \begin{cases} 1 & \text{if } L(i) \leq d < U(i) \\ 0 & \text{otherwise} \end{cases}$$

with

$$L(i) = i \times \frac{n}{N_b} \text{ and } U(i) = (i + 1) \times \frac{n}{N_b}$$

where $n$ is the number of possible descriptor values and $N_b$ is the number of histogram bins.

The resulting feature vector is defined as

$$H = \left\{ \frac{H^0}{(W_S)^2}, \frac{H^1}{(W_S)^2}, \ldots, \frac{H^{m-1}}{(W_S)^2} \right\} \tag{7}$$

where $m$ is the number of sub-block[1].

[1] Note that the superscripts in $H^0, H^1, \ldots$ are not exponents but orders. Hence, $H^0$ denotes the first histogram, $H^1$ is the second histogram, and so on.

### Histogram of Oriented Gradients

The Histogram of Oriented Gradients (HOG) descriptor was first proposed as a local-histogram descriptor for human detection in *Dalal & Triggs (2005)* and since then has been successfully applied to facial expression recognition (*Chen et al., 2014*; *Kumar, Happy & Routray, 2016*; *Hussein et al., 2022*). In this study, Scikit-image's implementation https://scikit-image.org/docs/dev/auto_examples/features_detection/plot_hog.html of the HOG descriptor was used. The HOG descriptor is calculated as follows. The first stage computes first-order image gradients by differentiating along the $x$ and $y$ directions. In the second stage, the first-order gradients are used to calculate the magnitude and orientation of the gradient images. In the third stage, the images are divided into cells, and a histogram is computed for each cell. Each cell histogram divides the gradient orientation range into a fixed number of intervals (or bins), and the gradient magnitudes of the pixels in the cell are used to vote into the histogram (see more details in the next paragraph). In the fourth stage, the cells are grouped into larger spatially connected groups called blocks, and block descriptors are calculated. A block descriptor is created by concatenating the cell histograms in that block and normalizing them (further explained below). Note that the blocks overlap, so cells are shared between blocks and appear more than once (but with

different normalizations) in the final feature vector. Finally, all the block descriptors are concatenated to form the final HOG feature vector.

Consider $L(x, y)$ to be the intensity function of an $M \times N$ gray-scale image. The first order image gradients along the $x$ and $y$ directions are defined as at coordinate $(x, y)$ are defined as

$$I_x = L(x - 1, y) - L(x + 1, y) \tag{8}$$
$$I_y = L(x, y - 1) - L(x, y + 1). \tag{9}$$

Given the differential components, $I_x$ and $I_y$, the gradient magnitudes and orientations of the image are defined as

$$M(x, y) = \sqrt{I_x^2 + I_y^2} \tag{10}$$

$$\alpha = \frac{180}{\pi} (\arctan2(I_y, I_x) \bmod \pi) \tag{11}$$

where arctan2 is the four-quadrant inverse tangent, which yields values between $-\pi$ and $\pi$. Using the gradients magnitude and orientation, the histogram of oriented gradients for each cell are defined as

$$H_{cell}(i, j, k) = \sum_{(u,v) \in C_{i,j}} M(u, v) \ if \ \alpha(u, v) \in \theta_k \tag{12}$$

$$0 < i < \frac{M}{cellSize} - 1, \ 0 < j < \frac{N}{cellSize} - 1$$

where $C_{i,j}$ is a set of $x$ and $y$ coordinates within the $(i, j)$ cell and $\theta_k$ is the range of each bin e.g., for 6 orientations, $\theta_1 = [165°, 180°) \cup [0°, 15°)$, $\theta_2 = [15°, 45°)$, $\theta_3 = [45°, 75°)$, $\theta_4 = [75°, 105°)$, $\theta_5 = [105°, 135°)$ and $\theta_6 = [135°, 165°)$. Therefore, $H_{cell}(i, j, k)$ represents the histogram of oriented gradients at the $(i, j)$ cell. After obtaining cell histograms, the cells are grouped in blocks and normalized as follows:

- First, the block descriptor is built by concatenating cell histograms within the block:

$$H_{block} = \left( H_{cell}^1 H_{cell}^2 \cdots H_{cell}^{n_c} \right) \tag{13}$$

where $n_c$ is the number of cells per block. In this study $n_c = 3$.

- Normalize the descriptor using L2 normalization:

$$\widehat{h_l} = \frac{h_l}{\sqrt{\sum_l h_l^2 + e^2}} \tag{14}$$

where $h_l$ is the $l$-th element of the block descriptor, $\widehat{h_l}$ is the corresponding normalized value, and $e$ is a constant to prevent division by zero.

- Limit the maximum value to 0.2:

---

**Algorithm 1 Histogram of oriented gradients.**

**Inputs:**

$I$ : input image

**Outputs:**

$H_{hog}$ : feature vector

1: Compute first order image gradients using Eqs. (8) and (9).

2: Compute gradient orientations and gradient magnitudes using Eqs. (10) and (11)

3: Build the histograms of oriented gradients for all the cells using Eq. (12).

4: Build a descriptor for all the blocks using normalization Eqs. (13)–(15).

5: Return a feature vector by concatenating all the block descriptors Eq. (16).

---

$$h'_l = \begin{cases} 0.2 & \text{if } \hat{h}_l > 0.2 \\ \hat{h}_l & \text{otherwise} \end{cases} \qquad (15)$$

- Repeat L2 normalization using Eq. (14).

- Concatenate all the block descriptors to produce the final HOG feature vector:

$$H_{hog} = \left( H^1_{block} H^2_{block} \cdots H^{n_b}_{block} \right) \qquad (16)$$

where $n_b$ is the number of blocks.

Algorithm 1 summarizes the calculation of a HOG feature vector.

### Local Binary Pattern

The Local Binary Pattern descriptor was first introduced to describe textural appearance in pattern recognition problems. The LBP histogram contains information relating to the distribution of the local micro-patterns, such as flat areas and edges in an image. The original LBP descriptor is computed by considering each pixel of an input image $g_c$ as the 'threshold' of a $3 \times 3$ grid and assigning a value of 1 or 0 to each of the eight surrounding pixels. The following rule is assumed: if the neighboring pixel is less than the threshold it is assigned the value 0, otherwise 1 is assigned. A binary code is then read from the sequence of 0s and 1s and converted to a decimal number. The decimal number is then assigned to the center (or threshold) pixel. Once this operation has been performed on the entire image, the LBP image is obtained. The histogram of LBP image is then used as a feature vector (*Ojala, Pietikäinen & Mäenpää, 2000*). Figure 5 gives an example of computing the LBP code of a pixel in a $3 \times 3$ neighborhood block.

LBP can be extended to allow any number of neighbors in a circular neighborhood and bilinear interpolation of pixel values (*Ojala, Pietikainen & Maenpaa, 2002*). In this study, the LBP descriptor was implemented with three neighbors and a radius of 1. The new LBP function is defined as

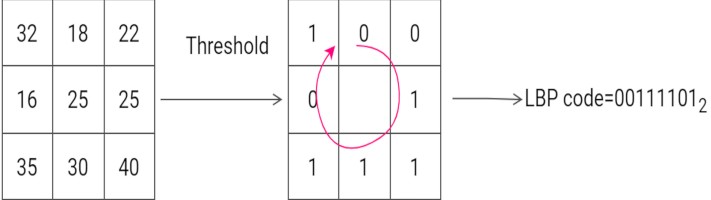

**Figure 5 An example of the computation of LBP on a 3 × 3 neighborhood. The threshold is 25 and the labels are read anticlockwise to produce a binary pattern of 0.** An example of the computation of LBP on a 3 × 3 neighborhood. The threshold is 25 and the labels are read anticlockwise to produce a binary pattern of 0.               

---

**Algorithm 2  Local binary patterns.**

**Input** Image: $I$

**Output** Feature vector: $(H^0 H^1 \cdots H^{N_b - 1})$

Divide the input image into blocks $(I_b)_{b=0,1,\ldots,N_b-1}$.

**for** each block, b **do**

    Compute the LBP codes $f_c^b$ for the block $b$ using Eq. 17.

    Compute the histogram $H^b$ using Eq. 18.

**end for**

Concatenate the histograms, $H = (H^b)_{b=0,1,\ldots,N_b-1}$.

---

$$f_c = \sum_{j=0}^{P-1} S(g_j - g_c) 2^j \tag{17}$$

where $1 \leq x \leq w, 1 \leq y \leq h$ and $S(A)$ is defined as

$$S(A) = \begin{cases} 1 & \text{if } A \geq 0 \\ 0 & \text{else} \end{cases}$$

and $g_j$ corresponds to the neighboring intensity values; $w$ and $h$ are the width and height of the input image respectively.

    After computing all the LBP codes, $f_c(x, y)$, a histogram, $(H_i)_{i=0,1,\ldots,n-1}$, is created that counts the occurrences of a range of LBP codes.

$$H_i = \sum_{x=0}^{w-1} \sum_{y=0}^{h-1} I(f_c(x, y) = i) \tag{18}$$

where $n$ is the number of bins in the histogram and $I(z)$ is defined as:

$$I(z) = \left\{ \begin{array}{ll} 1 & \text{if Z is true} \\ 0 & \text{else} \end{array} \right\}$$

Figure 6 shows the steps followed to compute LBP features of an image.

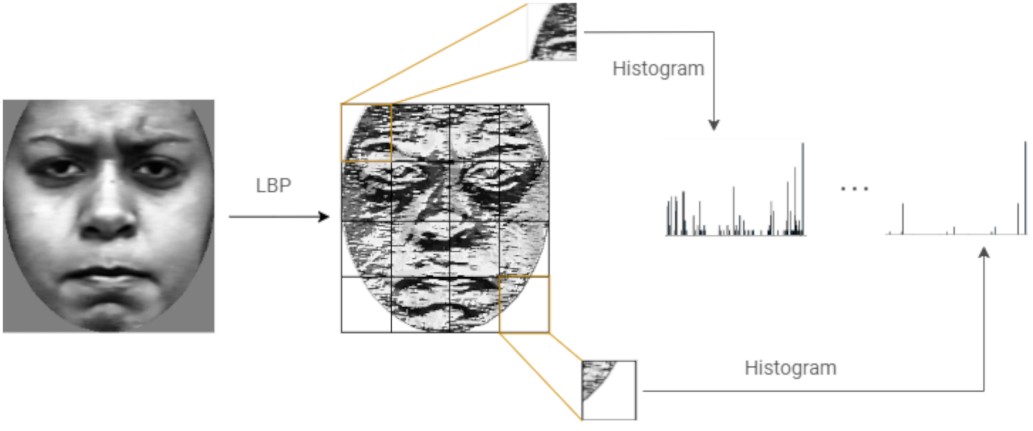

**Figure 6 Computing the LBP features of an image.**

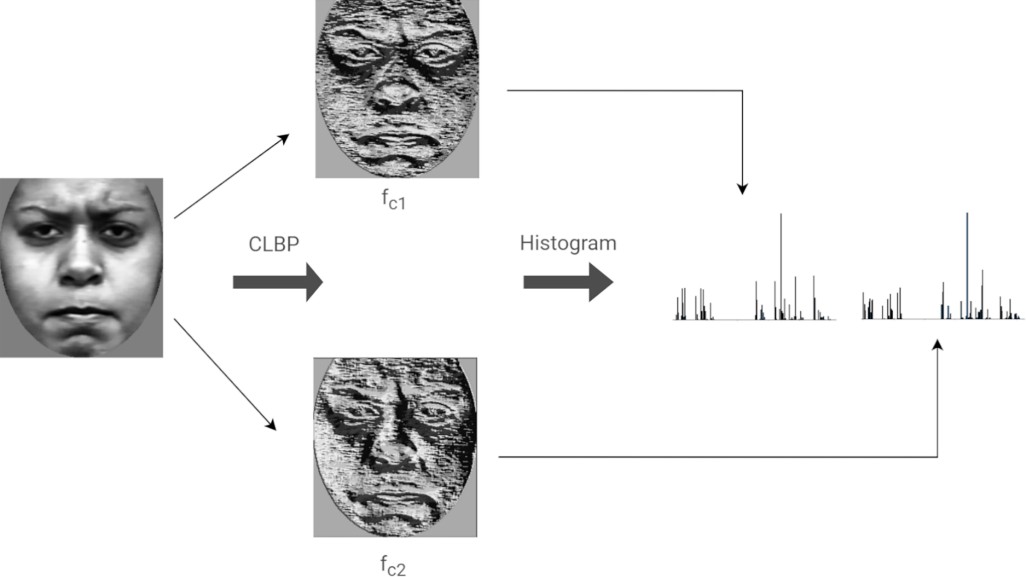

**Figure 7 Computing the LBP features of an image.**

### Compound local binary pattern

Compound Local Binary Patterns (CLBP) extends the LBP descriptor. CLBP differs from LBP in that it encodes both the magnitude as well as the sign of the differences between a center (or threshold) pixel and its $P$ neighbors. Each pixel is treated as a center having $P$ neighbors. Each neighbor is given a 2-bit code as follows: the first bit is set to 0 if the neighboring pixel is less than the center pixel and is set to 1 otherwise. The second bit is set to 1 if the magnitude of the difference between the neighboring pixel and the center pixel is greater than the threshold, $M_{avg}$, otherwise, the bit is set to 0. The threshold value is defined as the average magnitude of the differences between the center pixel and its $P$ neighbors. The rest of this discussion considers $P = 8$, since the same is used in our implementation. Figure 7 illustrates the computation of CLBP features of an image.

---

**Algorithm 3** Compound local binary patterns.

**Input** Image: $I$

**Output** Feature vector: $H = (H^0 H^1 \dots H^{N_b - 1})$

Divide the input image into blocks $(I_b)_{b=0,1,\cdots,N_b-1}$.

**for** each block, b **do**

    Compute the sub-CLBP codes using Eqs. (20) and (21), respectively.

    Create a histogram for the codes, $H_1^b$ and $H_2^b$.

    Concatenate the sub-histograms, $H^b = \{H_1^b H_2^b\}$.

**end for**

Concatenate the 1D histograms, $H = \{H^b\}_{b=0,1,\dots,N_b-1}$

---

Let $f_c(x, y)$ be the function representing the CLBP code of a pixel positioned at $(x, y)$ defined as:

$$f_c(x, y) = \sum_{p=0}^{P-1} s(i_p, i_c) 2^{2p} \tag{19}$$

where $s(i_p, i_c)$ is defined as

$$s(i_p, i_c) = \begin{cases} 00 & \text{if } i_p - i_c < 0 \text{ and } |i_p - i_c| \leq M_{avg} \\ 01 & \text{if } i_p - i_c < 0 \text{ and } |i_p - i_c| > M_{avg} \\ 10 & \text{if } i_p - i_c \geq 0 \text{ and } |i_p - i_c| \leq M_{avg} \\ 11 & \text{otherwise} \end{cases}$$

$i_c$ and $i_p$ represent the gray values of the center $c$, and a neighbor $p$, and $M_{avg}$ is the average magnitude of the differences between $i_c, i_p$ pairs in the local neighborhood.

The CLBP binary code is further subdivided into two sub-CLBP patterns $f_{c1}$ and $f_{c2}$. Hence, during implementation, Eq. (19) is not used but instead Eqs. (20) and (21) are used. The sub-CLBP patterns can be defined as follows:

$$f_{c1} = \sum_{p \in O, k \in E} \left( S_{1p}(i_p, i_c) 2^{k+1} + S_{2p}(i_p, i_c) 2^k \right) \tag{20}$$

$$f_{c2} = \sum_{p \in O, k \in E} \left( S_{1p}(i_p, i_c) 2^{k+1} + S_{2p}(i_p, i_c) 2^k \right) \tag{21}$$

where $O = \{1, 3, 5, 7\}$, $E = \{0, 2, 4, 6\}$ and $S_{1p}(i_p, i_c)$ and $S_{2p}(i_p, i_c)$ are defined as

$$S_{1p}(i_p, i_c) = \begin{cases} 0 & \text{if } i_p - i_c < 0 \\ 1 & \text{if } i_p - i_c \geq 0 \end{cases}, \quad S_{2p}(i_p, i_c) = \begin{cases} 0 & \text{if } |i_p - i_c| \leq M_{avg} \\ 1 & \text{if } |i_p - i_c| > M_{avg} \end{cases}$$

with

$$M_{avg} = \frac{1}{8} \sum_{p=0}^{7} |i_p - i_c|$$

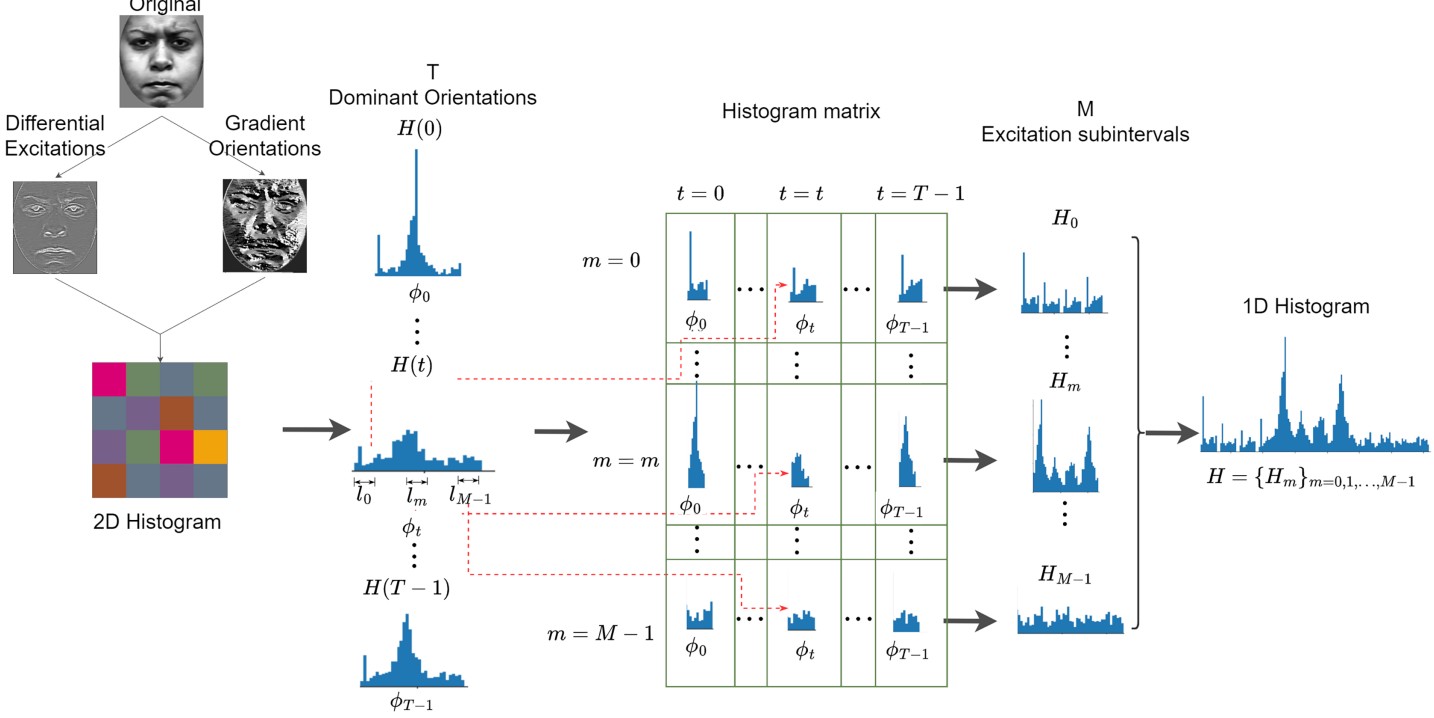

**Figure 8  Extraction of WLD features from an input image.**

### Weber's Local Descriptor

Weber's Local Descriptor (WLD) is inspired by the human perception of a pattern, which depends not only on the change of stimulus (such as sound and lighting) but also on the original intensity of the stimulus (*Chen et al., 2009*). WLD is made of two components: a differential excitation ($\xi$) and a gradient orientation ($\theta$). The differential component is expressed as the ratio between two terms: the first is the relative intensity differences of a current pixel against its neighbors; the other is the intensity of the current pixel. The second component of WLD is the gradient orientation. A feature vector is generated by first rearranging the differential excitations into subgroups that are used to form histograms, following a particular rule that is further discussed below. Histograms are ultimately reordered and concatenated to form a single feature vector. Figure 8 is an illustration of how WLD feature vectors are extracted from an input image. Algorithm 4 summarized the extraction of WLD features.

Consider a $3 \times 3$ pixels block $G_s$ taken from an input image $G$, the WLD operator can be applied as follows:

- Compute the differential excitation component which is expressed as:

$$\xi(x_c) = \tan^{-1}\left(\frac{V_s^{00}}{V_s^{01}}\right) = \tan^{-1}\left[\sum_{i=0}^{p-1}\left(\frac{x_i - x_c}{x_c}\right)\right] \tag{22}$$

---

**Algorithm 4** Weber's Local Descriptor.

**Input** Image: $I$

**Output** Feature vector: $H = (H^0 H^1 \cdots H^{N_b-1})$

Divide the input image into blocks $(I_b)_{b=0,1,\cdots,N_b-1}$.

**for** each block, b **do**

    Compute the differential excitations, $\xi_j$ using Eq. (22).

    Compute the gradient orientations, $\theta_j$ using Eq. (23).

    Map the gradient orientation, $\theta'_j$ using Eq. (24).

    Quantize the mapped orientations into $T$ dominant orientations, producing $\Phi_t$ using Eq. (25).

    Create sub-histograms $H_{m,t}$ using Eq. (26)

    Concatenate the sub-histograms $H^b = \{H_{m,t}\}_{t=0,1,...,T-1;m=0,1,...,M-1}$

**end for**

Concatenate the 1D histograms, $H = \{H^b\}_{b=0,1,...,N_b-1}$

---

- The gradient orientation $\theta(x_c)$ which is the second component of WLD is computed as follows:

$$\theta(x_c) = \tan^{-1}\left[\frac{V_s^{11}}{V_s^{10}}\right] = \tan^{-1}\left[\frac{x_5 - x_1}{x_7 - x_3}\right] \tag{23}$$

$$G_s = \begin{bmatrix} x_0 & x_1 & x_2 \\ x_7 & c & x_3 \\ x_6 & x_5 & x_4 \end{bmatrix}$$

$$\xi(x_c) \in \left[-\frac{\pi}{2}, \frac{\pi}{2}\right] \quad \theta(x_c) \in \left[-\frac{\pi}{2}, \frac{\pi}{2}\right]$$

- Map the gradient orientations from the interval $\left[-\frac{\pi}{2}, \frac{\pi}{2}\right]$ to the interval $[0, 2\pi]$ by applying the following function:

$$\theta' = \arctan 2(V_s^{11}, V_s^{10}) \tag{24}$$

$$\arctan 2(V_s^{11}, V_s^{10}) = \begin{cases} \theta & \text{if } V_s^{11} > 0 \text{ and } V_s^{10} > 0 \\ \pi - \theta & \text{if } V_s^{11} > 0 \text{ and } V_s^{10} < 0 \\ \theta - \pi & \text{if } V_s^{11} < 0 \text{ and } V_s^{10} < 0 \\ 2\pi + \theta & \text{if } V_s^{11} < 0 \text{ and } V_s^{10} > 0 \end{cases}$$

$$V_s^{11} = x_5 - x_1; \quad V_s^{10} = x_7 - x_3$$

- The gradients $\theta'$ are further quantized into $T$ dominant orientations using the function.

$$\Phi_t = f_q(\theta') = \frac{2t}{T}\pi \tag{25}$$

$$t = \mathrm{mod}\left(\left\lfloor \frac{\theta'}{2\pi/\mathrm{T}} \right\rfloor, \mathrm{T}\right)$$

where $t$ is a linear function that maps a differential excitation with orientation $\theta'$ to a particular dominant orientation $\Phi_t$. Next, a 2D histogram $\left(\mathrm{WLD}(\xi_j, \Phi_t)\right)_{j=0,1,\ldots,N-1,\,t=0,1,\ldots,T-1}$ is created. $N$ is the dimensionality (number of pixels) in an image and $T$ is the number of dominant orientations. The 2D histogram has $C$ rows and $T$ columns, where $C$ is the number of cells in each orientation. The value of $C$ is explained in the subsequent part of this section. Intuitively, each cell of the 2D histogram represents the frequency of a differential excitation interval on a dominant orientation.

The 2D histogram is further encoded into a 1D histogram, H as follows. First, each column of the 2D histogram is projected to form a 1D histogram, $(H(t))_{t=0,1,\ldots,T-1}$. Each histogram is produced by grouping differential excitations $\xi_j$ corresponding to a dominant orientation $\Phi_t$.

Next, each sub-histogram, $H(t)$, is divided into $M$ segments $\left(H_{m,t}\right)_{m=0,1,\ldots,M-1}$. The sub-histograms $H_{m,t}$ collectively form a histogram matrix. Each row of this matrix corresponds to a differential excitation segment and each column corresponds to a dominant orientation. The histogram matrix is reorganized as a 1D histogram by first concatenating the rows of the histogram matrix as a histogram, $H_m = \{H_{m,t}\}_{t=0,1,\ldots,T-1}$. The resulting M sub-histograms are concatenated as a 1D histogram, $H = \{H_m\}_{m=0,1,\ldots,M-1}$.

Note, that each sub-histogram, $H(t)$, is divided into $M$ segments, $H_{m,t}$, by dividing the interval of differential excitations $l = \left[-\frac{\pi}{2}, \frac{\pi}{2}\right]$ into $M$ segments $l_m$. Furthermore, each sub-histogram $H_{m,t}$ is an $S$-bin histogram defined as:

$$H_{m,t} = \{h_{m,t,s}\} \tag{26}$$

$$h_{m,t,s} = \sum_j \delta(S_j = s)$$

where $S_j$ and $\delta(A)$ are defined as

$$S_j = \mathrm{mod}\left(\left\lfloor \frac{\xi - \eta_{m,l}}{(\eta_{m,u} - \eta_{m,l})/S} \right\rfloor, S\right)$$

$$\delta(A) = \begin{cases} 1 & \text{if } A \text{ is true} \\ 0 & \text{otherwise} \end{cases}$$

with $\eta_{m,l} = \left(\frac{m}{M} - \frac{1}{2}\right)\pi$ and $\eta_{m,u} = \left(\frac{m+1}{M} - \frac{1}{2}\right)\pi$

## Support vector machines

The support vector machine (SVM) is a machine learning algorithm whose objective is to learn how to classify input features by defining a separating hyperplane. Considering $l$ training feature vectors $x_1, x_2, \ldots, x_l$ with their corresponding labels $y_1, y_2, \ldots, y_l$, where $y_i \in \{-1, 1\}$ for $i = 1, 2, \ldots, l$; the following primal optimization problem can be defined as:

$$\min_{w,b,\xi} = \left\{ \frac{1}{2} w^T w + C \sum_{i=1}^{l} \xi_i \right\}. \tag{27}$$

Subject to

$$y_i(w^T \Phi(x_i) + b) \geq 1 + \xi_i$$
$$\xi_i > 0, i = 1, \cdots, l$$

where $\xi_i$ and $\xi$ are the classification error for the $i^{th}$ training vector, and the total classification error, respectively; $w$ is the normal vector to the hyperplane; $\frac{b}{||w||}$ represents the offset of the hyperplane from the origin along the normal vector $w$ ($|| \cdot ||$ being the norm operator); $\Phi(x_i)$ is the kernel function, that maps $x_i$ into a higher dimensional space and $C$ is the regularization parameter. The classical SVM approach requires only two classes. So, a set of SVMs and a one-against-one strategy can make support vector machines suitable for multi-class problems such facial expression recognition (*Knerr, Personnaz & Dreyfus, 1990*). The experiments discussed here use the multi-class SVM strategy. More precisely, scikit-learn's implementation of the LIBSVM library (https://scikit-learn.org/stable/modules/generated/sklearn.svm.SVC.html) was used.

To train an SVM classifier, a set of hyper-parameters must be selected: the kernel function, the regularization parameter (C), and the kernel coefficient (gamma). For each classifier, an original parameter optimization procedure is performed. Specifically, a grid-search cross-validation procedure is employed (*Hsu, Chang & Lin, 2003*). In grid-search cross-validation, all possible combinations of the parameters are used to train a classifier using 10-fold cross-validation. Then, the parameter values which produce the best average recall are selected. Table 2 gives the parameter selection for when each type of feature is used to train an SVM classifier.

# EXPERIMENTAL RESULTS AND DISCUSSION

## Experimental phase 1: the effect of face registration

This section studies the effect of face registration on the FER system. To achieve this, the system is first trained on features extracted from unregistered faces and the performance is recorded. Then, the system is trained on features of registered faces. Note than the non-registered images are resized to $65 \times 65$ pixels, while the registered images are resized to $59 \times 65$ pixels to strike a balance between performance and feature vector length. Also, the parameters used for feature extraction are those recorded in Table 2.

**Table 2 Calibration of the SVM training hyper-parameters.**

| Descriptor | C | Gamma | Kernel |
|---|---|---|---|
| HOG | 1,000 | 0.05 | RBF |
| LBP | 1,000 | 0.05 | RBF |
| CLBP | 1,000 | 0.05 | Linear |
| WLD | 1,000 | 0.05 | Linear |

**Table 3 Recognition performance per class before and after registration.**

| | HOG | | LBP | | CLBP | | WLD | |
|---|---|---|---|---|---|---|---|---|
| | NR | RE | NR | RE | NR | RE | NR | RE |
| Anger | 88.9 | 93.3 | 68.9 | 91.1 | 84.4 | 80 | 77.8 | 93.3 |
| Disgust | 91.5 | 96.6 | 94.9 | 93.2 | 83.1 | 94.1 | 93.2 | 94.9 |
| Fear | 92 | 100 | 90 | 100 | 96 | 98 | 92 | 98 |
| Happiness | 98.6 | 98.6 | 95.7 | 98.6 | 91.3 | 97.1 | 95.7 | 98.6 |
| Sadness | 96.4 | 98.2 | 91.1 | 96.4 | 91.1 | 96.4 | 98.2 | 98.2 |
| Surprise | 98.5 | 98.5 | 97.1 | 97.1 | 95.6 | 98.5 | 95.6 | 98.5 |
| Average | 94.1 | 97.4 | 89.3 | 96.1 | 90.1 | 94.1 | 92.2 | 96.9 |

**Note:**
RE, Registered; NR, Non-registered.

Table 3 shows each expression's difference in recognition rate, showing that registration either increases the recognition rate (*e.g.*, fear) or has no effect (*e.g.*, HOG happiness). The most significant increase is observed for the LBP descriptor for the anger expression. The results reveal that face registration leads to a better recognition rate regardless of the descriptor used. These results suggest that face registration has a positive (and non-negligible) impact on the feature extraction phase and, ultimately, on the recognition performance. Figure 9 shows an example of a registered image compared to a non-registered one.

## Experimental phase 2: optimizing spatial local descriptors

This section evaluates the performance of four local descriptors: HOG, LBP, CLBP, and WLD. Several classifiers are trained on features from different descriptors, and the average recalls during 10-fold cross-validation are recorded. The classifiers are trained on features extracted from two datasets: CK+ dataset of six expressions and RFD dataset of seven expressions. A parameter selection is performed on each descriptor to find the configuration which gives the best performance. There are two parameters for the extraction of LBP and CLBP features: the histogram bin size and the block size. Bin sizes of 8, 16 and 32, and block sizes within the range 2 to 24 (inclusive) are available for selection (where the block size refers to the side length of a square). For the HOG descriptor, the number of orientation bins and cell size are tuned. The orientations used are 3, 5, 7, 9, 12,

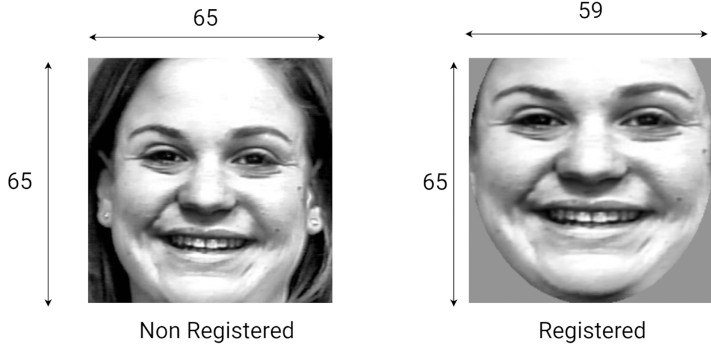

**Figure 9 Sample of non-registered and registered images.**

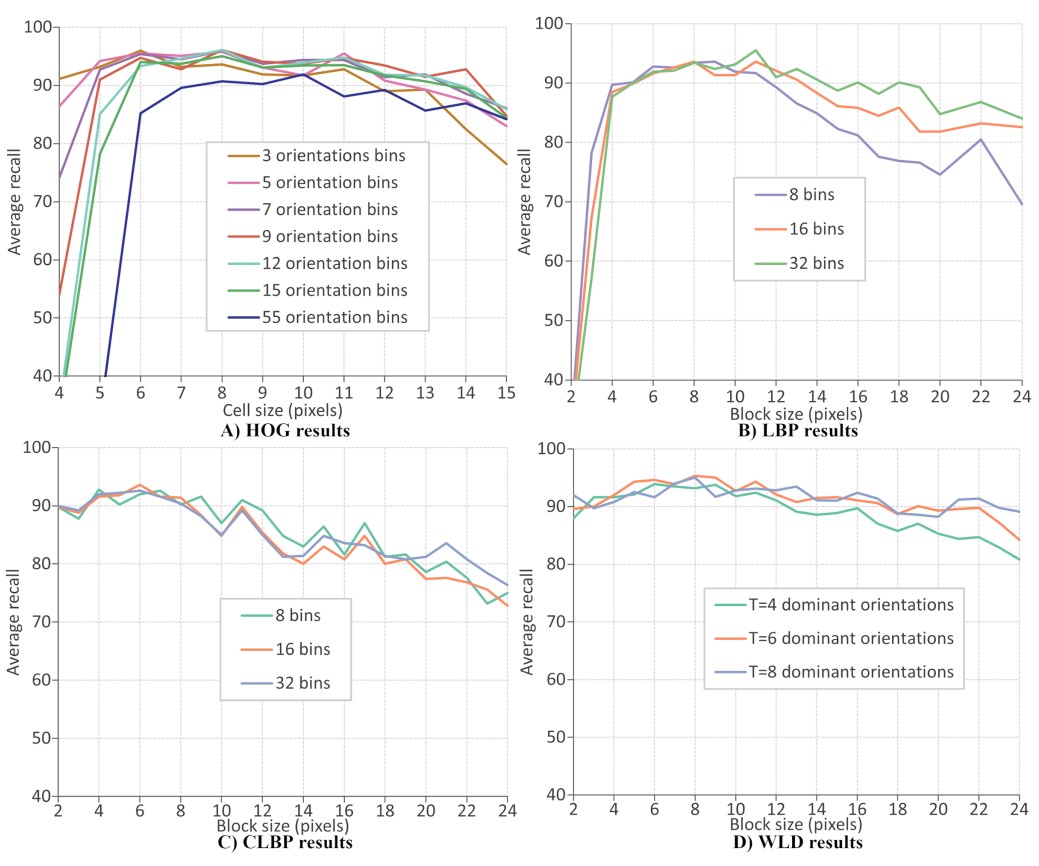

**Figure 10 LBP results.**               

15 and 55 and the cell size are varied between four and 15 (inclusive). Concerning the WLD descriptor, four parameters must be tuned (or optimized): *T*, *M*, *S*, and the block size. Block sizes ranging from 2 and 24 (inclusive) are available for selection while *T*, *M* and *S* can take values of 4, 6, and 8.

Figures 10A–10D give the FER results (expressed as average recall) when each descriptor is optimized on the CK+ dataset of six expressions. The best parameters for extracting HOG descriptor are reported as nine orientations and a cell size of 8, giving a

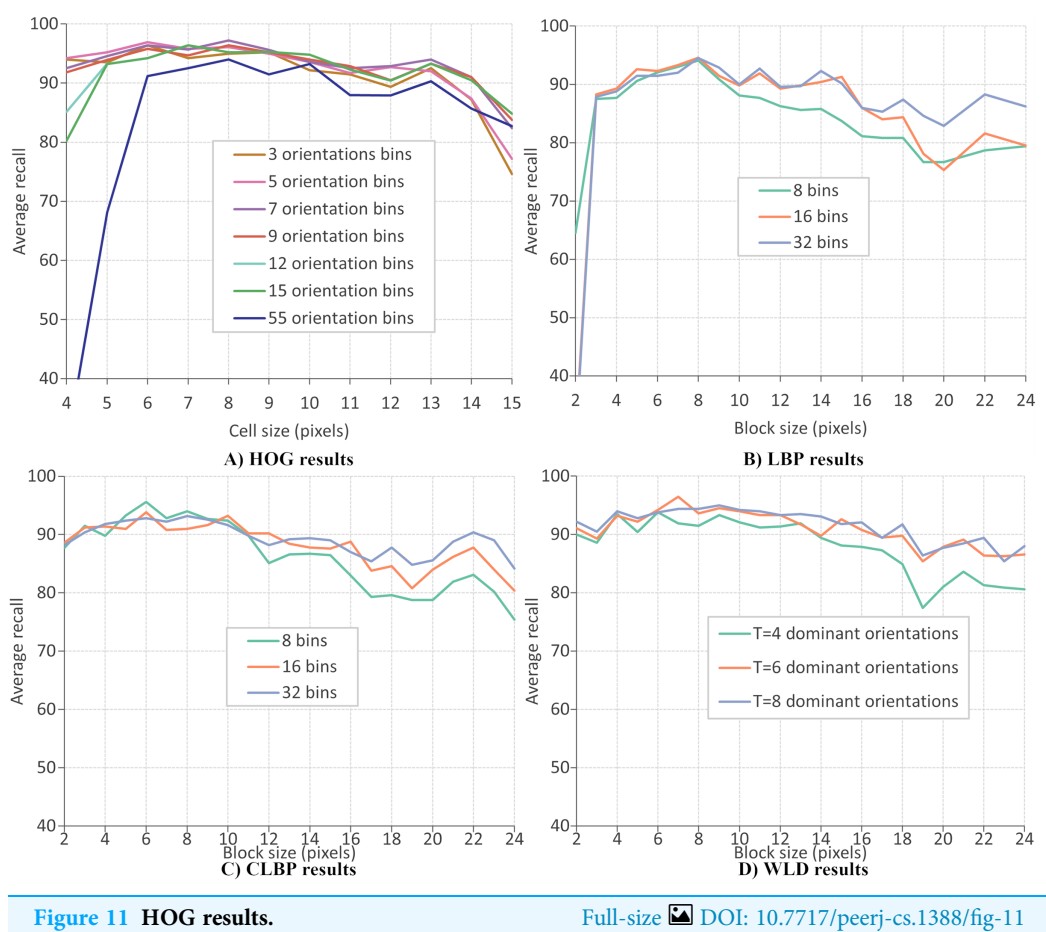

**Figure 11 HOG results.**     

recognition rate of 97.4%. Concerning the LBP features, the recognition rate has a maximum value of 96.1% when the number of bins is 16 and the block size of 7 pixels. The best recognition rate for the CLBP descriptor is 94.1% when the number of bins is eight, and the block size is 11. Finally, the FER results when using WLD features, where the best performance (96.9%) when the parameters are block size of 8, $T = 8$, $M = 4$, and $S = 4$. The same parameter optimization procedure is repeated; the features are extracted from the RFD dataset of six expressions. Figure 11A shows that the best parameters of HOG are 7 orientations and a cell size of 8 (RR = 97.2%). From Fig. 11b, the best LBP parameters are reported as 16 bins and a block size of 8 (RR = 94.6%). The best parameters for CLBP are 8 bins and 6-pixel blocks as shown in Fig. 11c (RR = 95.6%). Finally, Fig. 11d shows the best parameters of WLD as a block size of 7, $T = 6$, $M = 4$, and $S = 4$ (RR = 96.5%). The results show that the recognition rate rises and falls as the block size varies. This suggests that smaller blocks lead to too much fine-grain information, which is less useful in representing Action Units. On the other hand, larger blocks may merge multiple Action Units, which is less useful because it mixes discriminatory and non-discriminatory information. A similar finding was presented in *Moore & Bowden (2011)*.

The above results mean that the choice of parameter values improved local descriptors' performance. Tables 4 and 5 give the best parameter selection and feature vector length for

**Table 4  Best parameters for the extraction on the CK+ dataset.**

| Descriptor | Number of bins | Block size | T | M | S | Feature vector length |
|---|---|---|---|---|---|---|
| HOG | 9 | 8 | | | | 2,430 |
| LBP | 16 | 7 | | | | 1,440 |
| CLBP | 8 | 11 | | | | 576 |
| WLD | | 8 | 8 | 4 | 4 | 9,216 |

**Table 5  Best parameters for the extraction on the RFD dataset.**

| Descriptor | Number of bins | Block size | T | M | S | Feature vector length |
|---|---|---|---|---|---|---|
| HOG | 7 | 8 | | | | 1,890 |
| LBP | 16 | 8 | | | | 1,152 |
| CLBP | 8 | 6 | | | | 1,760 |
| WLD | | 7 | 6 | 4 | 4 | 8,640 |

each descriptor on each dataset. It can be observed that the most frequent value for block size is 8, and other values are close, *e.g.*, 6 and 7. This means that a block size of 8 × 8 pixels could be the best window for extracting features at the current resolution. On the other hand, the number of histogram bins is different for all descriptors.

## Comparison with the state-of-the-art

Comparing different FER systems in the literature is a challenging task because many studies omit or poorly discuss their evaluation strategy. Also, the training-testing data split often varies from one study to another. For this reason, only approaches evaluated using a 10-fold cross-validation method and the recognition rate (or average recall) were compared with this study. Table 6 gives various recent state-of-the-art approaches evaluated on the extended Cohn-Kanade (CK+) dataset (when a value is not available, it is replaced by '–'). All the approaches used SVM for classification, showing its superior performance, especially when working with small datasets. The proposed approach using the HOG descriptor achieved the best recognition rate, improving the results in *Carcagnì et al. (2015)* by a difference of 1.6%. We also note that the proposed approach gives a better recognition rate for LBP, CLBP, and WLD descriptors than previously obtained results. This improvement is probably due to this study's optimum selection of extraction parameters.

As observed in experimental results, the recognition rates first rise then fall as the block sizes are varied. Hence, the block size must be selected carefully when extracting each descriptor. Unlike previous studies, the current approach empirically determines the extraction parameters for LBP, CLBP, WLD, as opposed to determining them by trial and error. Another important aspect that makes the superiority of the proposed method is the histogram normalization. The HOG descriptor provides different histogram normalization methods which enhance its performance (*Dalal & Triggs, 2005*). Like

**Table 6 Performance comparison of our approach against various state-of-the-art approaches (CK+ 6 expressions).**

| Approach | Year | Feature | Feature vector length | Classifier | Face registration | No. images | Recognition rate (%) |
|---|---|---|---|---|---|---|---|
| *Kabir, Jabid & Chae (2012)* | 2012 | LDPv | 2,352 | SVM | No | 1,224 | 96.7 |
| *Carcagnì et al. (2015)* | 2015 | HOG | – | SVM | Yes | 347 | 95.8 |
| *Carcagnì et al. (2015)* | 2015 | LBP | – | SVM | Yes | 347 | 91.7 |
| *Carcagnì et al. (2015)* | 2015 | CLBP | – | SVM | Yes | 347 | 92.3 |
| *Carcagnì et al. (2015)* | 2015 | WLD | – | SVM | Yes | 347 | 86.5 |
| *Zhang et al. (2018)* | 2018 | CLGDNP | – | SVM | No | 1,482 | 95.3 |
| *Mandal et al. (2019)* | 2019 | DRADAP | – | SVM | No | 1,043 | 90.6 |
| *Hassan & Suandi (2019)* | 2019 | LBP | 2,478 | SVM | Yes | 150 | 96.2 |
| Proposed | 2022 | HOG | 2,430 | SVM | Yes | 347 | 97.4 |
| Proposed | 2022 | LBP | 1,440 | SVM | Yes | 347 | 96.1 |
| Proposed | 2022 | CLBP | 576 | SVM | Yes | 347 | 94.1 |
| Proposed | 2022 | WLD | 9,216 | SVM | Yes | 347 | 96.9 |

previous methods, the current approach used a technique called L2-Hys normalization which has proven to be superior than the others. However, the better performance shown by this approach could be attributed to the choice of the number of cells per block ($n_c$) used during the normalization process. Different $n_c$ values have shown to produce a measurable impact on the performance of HOG, and in this study $n_c = 3$ gave the best results. The other local descriptors were previously not normalized. Hence, we introduced a simple normalization method [2] which also enhanced their performances. This is evident because the results obtained with normalization were better than those obtained without normalization.

### Factors affecting facial expression recognition performance

The performance of the system is seriously affected by three main factors: the selection of datasets, the face registration, and the feature extraction parameters. Incorrect labeling also tends to introduce ambiguity between expressions and so can bring down the accuracy of predictions. The registration step is a crucial component to increase the classification performance. A correctly registered frontal face image must have all the facial components to guarantee extraction of the most key features and high prediction accuracy. This finding was pointed in *Carcagnì et al. (2015)* and our results confirm that the recognition performance is higher when using registered images than when the images are not registered.

### CONCLUSION

This article presented a novel method for optimized facial expression recognition. The proposed approach recognizes facial expressions in three stages: (1) face detection and registration, (2) feature extraction, and (3) classification. It is discovered that face registration is necessary to improve the performance of a FER system. The study has been validated on two popular facial expressions datasets, and the performance obtained is

---

[2] More details can be found in the "Feature Extraction" section.

comparable to the state-of-the-art. Four local descriptors were optimized, resulting in improved recognition rates. The actual values of the best parameters differed depending on the dataset used. However, it is worth mentioning that the best block sizes were very similar, taking values close to $8 \times 8$ pixels. Though the objectives of this study were achieved successfully, there is still room for improvement. The proposed FER system is limited to processing only frontal view faces. Future work must study the recognition of expressions on multi-view face datasets such as the BU-3DFE dataset (*Eroglu Erdem, Turan & Aydin, 2015*). Though the recognition rates were very high, this was at the expense of the feature vector length. Hence, future studies must explore feature space reduction techniques, such as AdaBoost, principle component analysis, and linear discriminant analysis, that can reduce the size of the feature vector while achieving similar or even better recognition rates.

## ABBREVIATIONS

| Notations | Description |
|---|---|
| CK+ | Extended Cohn Kanade |
| CLBP | Compound Local Binary Patterns |
| CLGDNP | Compact Local Gabor Directional Number Pattern |
| DRADAP | Decoder Regional Adaptive Affinitive Patterns |
| FER | Facial Expression Recognition |
| HOG | Histogram of Oriented Gradients |
| LBP | Local Binary Patterns |
| LDP | Local Directional Pattern |
| LDPv | Local Directional Patterns with variance |
| RBF | Radial Basis Function |
| RFD | Radboud Faces Database |
| SVM | Support Vector Machines |
| WLD | Weber's Local Descriptor |

### Funding
The authors received no funding for this work.

### Competing Interests
The authors declare that they have no competing interests.

### Author Contributions
- Antoine Badi Mame conceived and designed the experiments, performed the experiments, analyzed the data, performed the computation work, prepared figures and/or tables, authored or reviewed drafts of the article, and approved the final draft.
- Jules-Raymond Tapamo conceived and designed the experiments, analyzed the data, authored or reviewed drafts of the article, and approved the final draft.

## Data Availability

The Radboud Faces Database is available at:

https://rafd.socsci.ru.nl/RaFD2/RaFD?p=main.

The CK+ database is available at:

https://paperswithcode.com/dataset/ck#:~:text=CK%2B%20(Extended%20Cohn%
2DKanade%20dataset)&text=sadness%2C%20and%20surprise.-,The%20CK%2B%
20database%20is%20widely%20regarded%20as%20the%20most%20extensively,of%
20facial%20expression%20classification%20methods.

## Supplemental Information

Supplemental information for this article can be found online at http://dx.doi.org/10.7717/
peerj-cs.1388#supplemental-information.

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
