# Peer review of "Parameter optimization of histogram-based local descriptors for facial expression recognition"

_PeerJ Computer Science, doi:10.7717/peerj-cs.1388_

## Round 0.1 · original submission · Minor Revisions

This paper needs more explanatory description.

Reviewer 1 ·

Basic reporting

In this paper, a novel parameter optimization method of histogram-based local descriptors for facial expression recognition is proposed. The experimental results demonstrate the superiority and robustness of this algorithm. However, there are some unclear details, and the originality is not enough. Some comments may help improve the quality of the paper.
1) The author uses a lot of abbreviations. It is recommended to create a list of abbreviations to facilitate reading and understanding.
2) Section 2 reviews some background and related work on facial expression recognition. These methods are all important foundations for the proposed algorithms. In order to better highlight the effectiveness of this research work, it is suggested that the main contributions of this paper are given in numbered order at the end of this section.
3) In the experimental analysis part, the comparative analysis between the newly proposed algorithm and the previous algorithm (HOG, LBP, CLBP, WLD), it is recommended to conduct a detailed analysis to highlight the difference between the two parts and the innovation of this research.
4) The scale font size of the horizontal and vertical axes of some graphs in the experiment is too small, it is recommended to adjust.
5) The authors may miss the related latest work, such as [1]. Decoupling facial motion features and identity features for micro-expression recognition. PeerJ Computer Science. vol: 8, no: e1140; [2] Tackling Micro-Expression Data Shortage via Dataset Alignment and Active Learning. IEEE Transactions on Multimedia. DOI (identifier) 10.1109/TMM.2022.3192727. Please add the references.

Experimental design

no comment

Validity of the findings

no comment

Additional comments

no comment

Reviewer 2 ·

Basic reporting

Proposed method is to investigate four local descriptors to demonstrate them for facial expression recognition and highlights the importance of face registration. The manuscript is written easily understand. Here are some comments,
1. Many figures include texts that are too small to read them. It is recommended to enlarge texts.

Experimental design

If possible, the manuscript needs to include the following critical things to make it more concrete.
1. There are a lot of handicraft descriptors in this study. Are there any special reasons to pick the four descriptors (HOG, LBP, CLBP, WLD)?
2. Eye detection is on the step of face registration. How does the registration handle the issue if the eyes are not present clearly, such as occlusion?

Validity of the findings

The manuscript includes experimental results with the best parameter options for the descriptors. Quantitative results show the options work well for the target datasets (CK+ and RFD).

---

## Round 0.2 · accepted · Accept

Thank you for addressing all the reviewers' comments, I agreed with you in your response to the reviewer, the references, are not relevant to your manuscript.